# School Entry Vaccination Checks Allow Mapping of Under-Vaccinated Children in Zambia

**DOI:** 10.3390/vaccines13090924

**Published:** 2025-08-29

**Authors:** Megan P. Powell, Webster Mufwambi, Alvira Z. Hasan, Aliness M. Dombola, Christine Prosperi, Rodgers Sakala, Kelvin Kapungu, Gershom Chongwe, Prachi Singh, Qiulin Wang, Stella Chewe, Francis D. Mwansa, Constance Sakala, Elicah Kamiji, Patricia Bobo, Kennedy Matanda, Joan Manda, Amy K. Winter, Molly Sauer, Andrea C. Carcelen, Shaun A. Truelove, William J. Moss, Simon Mutembo

**Affiliations:** 1International Vaccine Access Center, Department of International Health, Johns Hopkins Bloomberg School of Public Health, Baltimore, MD 21231, USA; megan.p.powell1@gmail.com (M.P.P.); ahasan7@jhmi.edu (A.Z.H.); cprospe1@jhu.edu (C.P.); prachi@jhu.edu (P.S.); qwang120@alumni.jh.edu (Q.W.); msauer3@jhu.edu (M.S.); acarcel1@jhmi.edu (A.C.C.); shauntruelove@jhu.edu (S.A.T.); wmoss1@jhu.edu (W.J.M.); 2Department of Epidemiology, Johns Hopkins Bloomberg School of Public Health, Baltimore, MD 21205, USA; 3National Health Research and Training Institute, Lusaka 71769, Zambia; mufwambiwebster@gmail.com (W.M.); amdombola1987@gmail.com (A.M.D.); sakalarodgers@gmail.com (R.S.); kuchona@gmail.com (K.K.); gchongwe@gmail.com (G.C.); 4Ministry of Education, Government of Republic of Zambia, Lusaka 10101, Zambia; stellachewe@gmail.com; 5Ministry of Health, Government of Republic of Zambia, Lusaka 10101, Zambia; fmdien@gmail.com (F.D.M.); constancesakala@gmail.com (C.S.); kandinda2015@gmail.com (E.K.); drbobopm@gmail.com (P.B.); matandakennedy41@gmail.com (K.M.); joan31manda@gmail.com (J.M.); 6Center for the Ecology of Infectious Diseases, Department of Epidemiology and Biostatistics, University of Georgia, Athens, GA 30602, USA; awinter@uga.edu

**Keywords:** geographical information systems, immunization, health systems

## Abstract

Background: Geographic information systems (GIS) are a promising tool for mapping vaccination coverage and identifying missed communities, yet their use in low- and middle-income countries (LMICs) remains limited. In settings without standardized addresses such as schools or outreach sites, innovative methods are needed to collect and analyse spatial data. Schools offer a unique platform for identifying under-vaccinated children missed by routine or campaign efforts. Methods: During a pilot school vaccination screening program in Zambia, GIS reference maps of health facility catchment areas were developed from hand-drawn sketch maps, catchment area shapefiles, and coordinates of prominent landmarks. These maps were iteratively refined with input from local health staff. In caregiver interviews, data collectors used the maps to identify the child’s zone of residence within the health facility catchment area. Vaccination status was extracted from paper registries used during screening. Geographic heat maps were generated in ArcGIS to visualize under-vaccination by zone. Results: Of 535 children screened across 25 zones, 29% were under-vaccinated. Under-vaccination varied by zone, with clusters of missed children identified, for example, 50% of children in Kabushi Zone 6 were under-vaccinated, compared with much lower rates elsewhere. Conclusions: Pairing school-based vaccination checks with GIS mapping offers a scalable approach to identifying missed communities in LMICs. This method enables spatial analysis without household visits, supporting targeted immunization planning where traditional data systems fall short. However, because the study was limited to children enrolled in five purposively selected schools, out-of-school children and those in other schools were not represented. This selection bias may underestimate the true extent of under-vaccination, and future evaluations should incorporate broader and more representative populations.

## 1. Introduction

Over the past five decades, immunization programmes have averted 154 million deaths, largely driven by measles vaccination [1]. Despite these gains, vaccine-preventable diseases (VPDs) remain a major cause of childhood morbidity and mortality in low- and middle-income countries (LMICs), where access to immunization services faces several challenges [2]. In 2023 alone, 14.5 million children did not receive a single routine vaccine dose and another 6.5 million children were only partially immunized [3]. As is the case in other LMICs, VPDs in Zambia remain a public health concern, with periodic measles outbreaks occurring even among school-going children between 5 to 10 years [4,5]. In 2023 national coverage for the first dose of measles-containing vaccine was 80.3%, while the second dose lagged at 60.5% [6]. These vaccination coverages are below the measles herd immunity threshold of 95% [7]. While the Ministry of Health (MoH) conducts measles and rubella (MR) supplemental immunization activities (SIAs) to close immunity gaps, these SIAs primarily target younger children below the age of 5 years and miss school aged children, who are usually older than 5 years [8].

Un- and under-vaccination is shaped by multiple factors, including health system limitations, geographic and socioeconomic inequities, caregiver knowledge and attitudes, conflict or displacement compounded by lack of mechanisms for identifying communities which are likely to be missing doses [9]. Clustering of un- or under-vaccinated children can lead to an increased risk of VPD outbreaks [8,10,11,12,13]. Identifying communities with high concentrations of these children is imperative for targeting immunization efforts and efficiently reducing outbreak risk.

The school setting provides a platform where un- and under-vaccinated children can be identified. Mapping household data from un- and under-vaccinated children identified in schools can help program managers pinpoint geographic areas with high concentrations of under-vaccinated children. Such an approach can potentially be implemented in Zambia, where the School Health and Nutrition Policy (SHN) recommends vaccination checks at school entry [14]. However, this SHN policy is not currently implemented, and the country currently lacks a formal catch-up vaccination policy for children 5 years and above.

Geospatial maps can offer a simple spatial visualization of program resources, population distribution, and vaccination coverage to inform decision-makers on optimal service delivery strategies [15]. Geographical information systems (GIS) involve the use of computer software to both display and analyse spatial relationships in the form of maps [16]. Both UNICEF and the Vaccine Alliance (Gavi) have identified GIS as a potentially effective tool for mapping vaccination coverage and identifying missed communities, though its use is limited in LMICs [16,17]. This is largely due to the resource-intensive methodologies, infrastructure limitations, and technological advancements necessary for geospatial data collection [16]. In LMICs, the challenges posed by incomplete vaccination registries and a lack of standardized street addresses highlight the need for innovative strategies to effectively collect vaccination and geospatial data. This is particularly relevant when data collection occurs in settings other than the home, where the population of interest may be present for various purposes even as residential GIS coordinates cannot be obtained, such as schools, health facilities, outreach vaccination sites, or markets. Identifying key challenges and leveraging lessons learned from pilot studies that implement GIS is critical for advancing the field and enhancing immunization strategies in LMICs.

This study used GIS to convert hand-drawn paper maps from select local health facilities in Ndola District, Zambia, into digital representations of the health facilities’ catchment areas. Typically, health facilities depend on hand-drawn sketch maps for operational planning, which health workers are familiar with but often lack scale, completeness, accuracy, and critical information such as population density, infrastructure, and accessibility. Digital maps offer the chance to enhance accuracy and layer information that helps identify under-vaccinated communities and facilitates better resource allocation.

This report outlines the process of map development, data collection, and analysis, highlights challenges encountered and discusses how such methods can strengthen geospatial approaches for equitable vaccine delivery in LMICs.

## 2. Materials and Methods

### 2.1. Study Overview

The cross-sectional study was conducted across five purposively selected primary schools in Ndola District, Zambia, to identify geographical areas with a high prevalence of under-vaccinated children. In collaboration with the National Health Research and Training Institute (NHRTI, formerly the Tropical Disease Research Centre) in Zambia, children were screened for their vaccination status during school admissions from November 2022 through January 2023. The Zambian MoH organized school-based catch-up activities in February 2023 to vaccinate children identified as under-vaccinated during screening. Parents and caregivers could also have their child vaccinated at the health facility linked to the school. Follow-up interviews with parents and caregivers were conducted between April and June 2023 to collect household and demographic data, including geographic area of residence using reference maps.

### 2.2. Study Setting

Ndola is an urban district and the political capital of the Copperbelt Province. It houses 150 government and 68 private, church, or grant-aided primary schools linked to 25 primary health facilities. As of 2022, its population was 0.6 million (1). Study participants consisted of grade 1 children attending one of five primary schools: Masala, Kabushi, Chifubu, Mapalo, and Mackenzie school.

### 2.3. Vaccination Checks During School Admission

Vaccination status was collected by trained teachers and nurses using paper-based registers during school admission in selected schools. Parents or caregivers routinely submit an admission form along with an under-5 card or other documentation showing proof of age. This study involved an additional step where grade 1 teachers assessed the status of routine vaccination doses for oral polio (OPV), inactivated poliovirus (IPV), pentavalent (diphtheria-tetanus-pertussis [DTP]-containing vaccine), and measles-containing vaccine (MCV). Grade 1 teachers recorded the vaccination doses based on a child’s under-5 card. If a child did not have a card or if the card quality was poor, nurses completed the vaccination screening by interviewing a parent or caregiver with standard questioning. Teachers and nurses recorded each dose as either ‘yes’ (dose received) or ‘no’ (dose not received or unknown).

### 2.4. Reference Map Development and Implementation

The steps for reference map development are outlined in Figure 1. In Zambia, the Ministry of Education and MoH link each school to a nearby health facility (Table A1). Most health facilities use paper-based sketch maps, typically drawn by facility staff, to outline their catchment areas and key landmarks, and are often divided into smaller zones (Table A2). These paper sketch maps for each school’s linked facility were obtained from local health facilities. Baseline maps were generated using shapefiles of the district and health facility catchment area boundaries obtained from Akros Zambia. Using both the baseline and paper sketch maps, a zoned map was created in ArcGIS version 2.9, depicting the catchment area and zones. Guided by these zoned maps, the NHRTI team conducted field visits to identify prominent landmarks and collect geospatial coordinates.

The final reference maps incorporated these landmarks, clearly delineating zonal boundaries and key features of each catchment area. These maps were further piloted and refined based on feedback from the NHRTI team and health facility staff.

Data collectors were trained in how to use the reference maps and how to probe respondents to identify which zone their child lived in. Household information was collected through interviews at school. Interview data were collected in tablet-based electronic forms, and the data were uploaded to a central server. Vaccination status obtained at school admission was abstracted from the paper registries to the electronic system. The study team documented all of the procedures and lessons learned from implementing this project.

### 2.5. Statistical Analysis

After map development and implementation, a geographic heat map was generated using ArcGIS version 2.9 to display the prevalence of under-vaccinated children by zone in the reference map. Under-vaccination was defined as missing at least one of the routine vaccination doses (OPV, MCV, DTP-containing vaccine, IPV (if below 5 years)). In the dataset a missing dose was recorded as no, which could mean a child did not receive the dose or status was unknown. We treated children with unknown status as missing doses to ensure children who may have truly missed doses were not overlooked, in line with standard public health practice. Results were restricted to children enrolled at the linked school at the time of the study who were screened at admissions and whose caregivers participated in the follow-up interviews. Vaccination status for any zone with fewer than 10 children was excluded due to the small sample size. Analyses were conducted using R version 4.3.1 and ArcGIS version 2.9.

### 2.6. Ethical Considerations

The NHRTI Ethics Committee, Ndola, Zambia, the National Health Research Authority of Zambia, and the Johns Hopkins Bloomberg School of Public Health Institutional Review Board, Baltimore, USA, approved the protocol. Written informed consent from caregivers of children younger than 18 years and written assent from children between 10–17 years was obtained before participation in the interview.

## 3. Results

### 3.1. School Entry Vaccination Checks

A total of 535 children from across 25 health facility zones were enrolled in the school vaccination checks study. The median number of the enrolled children was 13 per zone (range: 0–133) (Table 1). Of these children, 29% (151) were identified as under-vaccinated. A total of 57 children (11%) were found to be living outside the catchment area. Only 20 caregivers/parents of children at Mackenzie School were interviewed, though they were excluded from the analysis due to the small sample size.

### 3.2. Mapping Under-Vaccinated Children

In settings without street addresses, identifying missed communities through school entry vaccination checks requires innovative methods of mapping communities. To develop the reference maps, those closely familiar with the area needed to be the central drivers in the mapping process. The maps were repeatedly piloted and refined based on feedback from the NHRTI team and the MoH facility staff. The paper sketch maps provided by the health facilities varied considerably in quality and readability. Specifically, the paper sketch maps from New Masala clinic, Mapalo clinic, and Chipokota Mayamba clinic featured clearly delineated zones, whereas the sketch map for Kabushi clinic lacked marked zones and Itawa clinic did not have a paper sketch map. Leveraging their local knowledge, the NHRTI team worked with the health facility staff to draw zone boundaries using roads, natural dividers, and neighbourhood lines within the Kabushi and Itawa clinics’ catchment areas. These paper sketch maps were subsequently digitized and refined using ArcGIS. Although the initial lack of quality paper sketch maps from two health facilities posed a challenge, collaborating with individuals possessing local knowledge of the area enabled us to effectively address and overcome these challenges.

Examining a specific example, Figure 2 outlines the process by which the reference map for Kabushi school was made. The paper sketch map did not have clearly defined zones but instead designated specific areas with arrows feeding into the clinic. Using the boundaries for the catchment area in the baseline maps, we first divided the area into 18 different zones following major roads and the areas defined in the paper sketch map. The resulting reference map was presented to the NHRTI team, who, with their local knowledge and in consultation with health facility staff, redrew the zone boundaries to have a total of seven zones.

While having more zones can lead to a greater level of precision in terms of where the respondent lives, it also makes it more challenging for data collectors to probe respondents on which zone the child lives in. This is because smaller zones may contain fewer recognizable landmarks and a higher likelihood of respondents identifying locations near zone boundaries, increasing the risk of misclassification and requiring more extensive probing. Furthermore, having a greater number of zones leads to a smaller sample size within each zone, making it more difficult to compare the percentage of under-vaccinated children between zones. For our study, it was most important to divide the catchment area in terms of its characteristics and commonalities, which requires that local knowledge such as understanding of neighbourhood names, commonly used landmarks, and natural boundaries be the driving force in the map’s development. We aimed to align the reference maps as closely as possible with the paper sketch maps generated by the local health facilities, as these paper sketch maps are intended to enable local healthcare workers to identify and address gaps in health services.

To make the reference maps user-friendly, the NHRTI team selectively chose the most prominent landmarks in the area and obtained their precise GIS coordinates to incorporate into the reference maps. Striking a balance in the number of landmarks displayed was crucial: too many could clutter the map, while too few might not offer sufficient reference points for parents to accurately identify their zones. Therefore, only the most significant landmarks were included, and the selection was refined during pilot testing with health facility staff. Figure A1 displays the final reference maps for all four schools included in the analysis. The reference maps were also accompanied by tables listing a more extensive number of landmarks alphabetically that was organized by zone to further assist data collectors during the interviews (Table A3, example for Kabushi school). The key lessons learned from developing the reference maps are provided in Table 2.

Training data collectors was crucial to the successful use of the reference maps. Data collectors underwent extensive training in how to probe respondents and use the reference map and a table of landmarks by zone to identify the zone in which the child resides (Table A3). This training was especially important as many respondents needed guidance to orient themselves to the reference maps. Figure 3 describes the process by which the data collector used the reference map and table, along with probing questions, to identify the zone in which the respondent lived. Data collectors practiced challenging scenarios and pilot tested the reference maps in the field to gain experience. They then held debriefing sessions to share successes and challenges and learn from each other.

### 3.3. Identifying and Mapping Missed Communities Through School Entry Vaccination Checks

Using the school vaccination screening data and the reference maps, we summarized the proportion of children under-vaccinated by zone. We observed that the proportion of children identified through the school screening as under-vaccinated varied by zone within catchment areas, identifying potential missed communities that were displayed by the geographic heat maps (Figure 4A–D). In Chifubu school, the proportion of under-vaccination was highest among children who reported living outside of the health facility catchment area (67%) (Figure 4C). The geographic heat maps for both Kabushi and Mapalo Schools most clearly identified those potential missed communities that had a higher percentage of under-vaccinated children relative to the rest of the catchment area (Figure 4A,B). For example, in Kabushi School, half of the children living in Zone 6 were under-vaccinated, a much higher percentage than in Zones 2, 3, and 4 (Figure 4A). These results suggest that those living in Zone 6 may reside in a missed community and be at greater risk of vaccine-preventable disease outbreaks compared with the other catchment area zones. For Chifubu and Masala schools, the proportion of under-vaccinated children was distributed relatively evenly among zones consisting of at least 10 children (Figure 4C,D).

## 4. Discussion

The spatial distribution of under-vaccinated children varied considerably across the catchment areas. In both Kabushi and Mapalo schools’ catchment areas, under-vaccinated children were disproportionately concentrated within a single zone (Figure 4A,B). In contrast, under-vaccination appeared to be evenly distributed across zones in Chifubu and Masala schools’ catchment areas (Figure 4C,D). These differences may reflect the relatively small sample sizes within each zone, and further research is needed to confirm these patterns. Nonetheless, the results indicate that the distribution of under-vaccinated children can vary substantially across settings. They suggest that a targeted vaccination strategy may be most effective in Kabushi and Mapalo, whereas such an approach may be less appropriate for Chifubu and Masala. Identifying areas and characterizing the distribution of under-vaccinated children is crucial for effectively targeting vaccination efforts.

This study demonstrates how school entry vaccination checks can be leveraged to identify and map missed communities using digitally transformed hand-drawn sketch maps that provide a clear spatial visualization of vaccination coverage across a catchment area, particularly where data collection is conducted outside the household setting. Our experience underscores the broader value of school entry vaccination checks and the importance of incorporating local knowledge, engaging community members, and tailoring data collection to local needs when mapping missed communities. Refining the reference maps through feedback from local nurses and pilot testing was critical to ensure the maps accurately reflect vaccination status within the catchment area [18].

Dougherty et al. employed a similar approach, using GIS to digitize health facility maps in northern Nigeria to enhance immunization planning [19]. Their study encountered several challenges, including inconsistent naming conventions for facilities and settlements, difficulties in estimating population sizes within mapped boundaries, and a shortage of local GIS expertise. Likewise, our study was unable to generate population estimates for the zones within the catchment area due to data gaps and a lack of standardization in the quality of paper sketch maps drawn by health facilities. Both our study and that of Dougherty et al. indicate that, while GIS mapping can be an effective decision-support tool for microplanning, further efforts are needed to standardize processes, build local capacity, and ensure data quality.

Previous studies have applied GIS technology in Zambia to identify and reach zero-dose children through household visits but not using school entry vaccination checks [17].

While household visits facilitate the collection of GIS coordinates, they are resource-intensive and difficult to scale up. In contrast, developing and utilizing reference maps during interviews for school entry vaccination checks presents a more scalable alternative to identifying under-vaccinated children in diverse settings, without the need for door-to-door visits [16].

In addition to school vaccination screenings, zoned GIS maps like the reference maps in this study could be used at health facilities or vaccination outreach sites to track patient residence, highlight missed communities, and inform microplanning for outreach services of vaccinations and other interventions. They can also be used to identify areas affected by infectious disease outbreaks among cases presenting to the health facility to inform response activities. Data-driven approaches to allocating resources are a priority for funders and governments and will become increasingly critical as systems are stretched due to changes in the funding landscape.

There is growing evidence that implementing GIS mapping is an effective tool for reaching missed communities and improving immunization programs [16]. For example, geospatial maps were used to guide polio elimination efforts across various countries, including Nigeria, Myanmar, and Cameroon. In certain settings evidence showed GIS technology improved the identification of zero-dose and under-immunized children though effective immunization targeting and microplanning efforts. Despite documented success, the use of GIS continues to be underutilized in LMICs, largely due to the long-term investment they require to develop and sustain [20]. It has also been noted that many GIS projects have not been shared publicly, and that there is a need to prioritize documenting lessons learned for the field to advance. Our study shed more light on how GIS mapping can potentially be used in a program such as school entry vaccination checks to identify communities missing public health services. It also provides more information on the challenges that are likely to be encountered in the use of GIS mapping in most LMICs.

A limitation of this mapping approach is the initial cost of GIS technology, and the limited availability of local GIS expertise [17]. In most LMIC software and equipment such as computers are expensive and are therefore not readily available. Developing digital maps requires substantial investments in training, dedicated time to develop skills and data must be regularly updated to maintain accuracy and utility. Although GIS mapping has been used in Zambia previous for vaccination programs, there is still limited in-country expertise because of GIS mapping has been conducted by external partners [17]. Strengthening local capacity is therefore critical to enable sustainable, locally led GIS mapping efforts. Zambia’s National Health Strategic Plan emphasizes the goal of enhancing health information systems through digital technologies to improve data quality and decision-making capabilities [21]. Gavi’s rapid review of GIS applications finds a consensus that GIS is a cost-effective tool to improve immunization coverage for zero-dose children, particularly in comparison to alternatives like national health surveys or door-to-door data collection [22]. The rapid review synthesized thirty-one studies examining the application of GIS mapping to identify and reach zero-dose children. Notably, most studies did not report cost estimates, underscoring the need for more information on the costs of GIS-based approaches [22]. For broader implementation, investments in local capacity building is essential, and the cost-effectiveness will ultimately depend on upfront resource allocation and long-term sustainability at the local level. Overall, GIS is considered a promising and yet underutilized tool for identifying underserved and missed communities.

There were several challenges and limitations to our mapping exercise. The most significant limitation was the study design, which relied on children enrolled in five purposively selected primary schools. As a result, children attending other schools within the health facility catchment area, as well as those not enrolled in school, were not represented. Because of this selection bias introduced by limited sampling of schools, we cannot report prevalence point estimates of zero dose in the identified communities. However, our study still demonstrates potential use of the school vaccination check in combination with GIS mapping to identify unvaccinated and under-vaccinated communities and shows the challenges which can be encountered with this approach. The use of the reference map to compare population characteristics between geographic areas relies on the ability to capture a representative study population and a sufficient sample size within each geographic area. Due to the study design, we could not capture children attending other schools within the health facility catchment area and children who did not attend school. Using data from school-based screenings to evaluate characteristics of the geographic settings where a child resides may be more effective in rural areas, where all schools within a health facility catchment area can be included in the screening, or where a single health facility serves a specified population. The number of children enrolled in our study was relatively small, especially when dividing each health facility’s catchment area into six or seven zones. These small numbers made it difficult to compare vaccination status across zones, and we had several zones marked as having few or no children residing in them. Given the small sample size, further investigation is needed to determine whether the observed patterns truly reflect missed communities or are due to sampling variation. Some trends in vaccination status and characteristics between zones may have been obscured due to the small sample size since we did not include all children in the catchment area. Therefore, our study could only compare the proportion of children who were under-vaccinated across zones, rather than total numbers. To avoid overinterpreting proportions due to small sample size, we excluded Mackenzie school and all zones with fewer than 10 children from the analysis. We also refrained from performing geospatial statistical analyses due to the small sample sizes within each zone.

School entry vaccination checks can not only identify under-vaccinated children but missed communities when linked to a mapping of the residences of those children. Innovative approaches for geospatial data collection are essential for identifying and reaching marginalized populations in LMICs, where data gaps and logistical challenges hinder traditional approaches. Through our lessons learned, this study advances GIS applications by developing a novel tool for collecting spatial data from children participating in a school entry vaccination program and without standardized addresses or GIS coordinates. Unlike traditional spatial analyses that rely on field-collected GIS data, this approach enables program managers to identify geographic clusters of under-vaccinated children and missed communities more efficiently, enhancing immunization planning and outreach.

## 5. Conclusions

School entry vaccination checks combined with GIS mapping offer a practical approach to identifying under-vaccinated children and missed communities in LMICs. By digitizing sketch maps and integrating local health worker input, this method provides clear spatial insights to support targeted immunization planning. While the study was limited by small sample sizes and the inclusion of only five purposively selected schools, the findings underscore both the promise and the challenges of applying GIS in settings with constrained data systems. Broader implementation will require investments in local GIS capacity and sustainable resource allocation to ensure accuracy and long-term utility. This study adds to growing evidence that school-based vaccination checks linked to GIS can strengthen immunization microplanning, improve the identification of underserved populations, and ultimately contribute to more equitable vaccination coverage.

## Figures and Tables

**Figure 1 vaccines-13-00924-f001:**
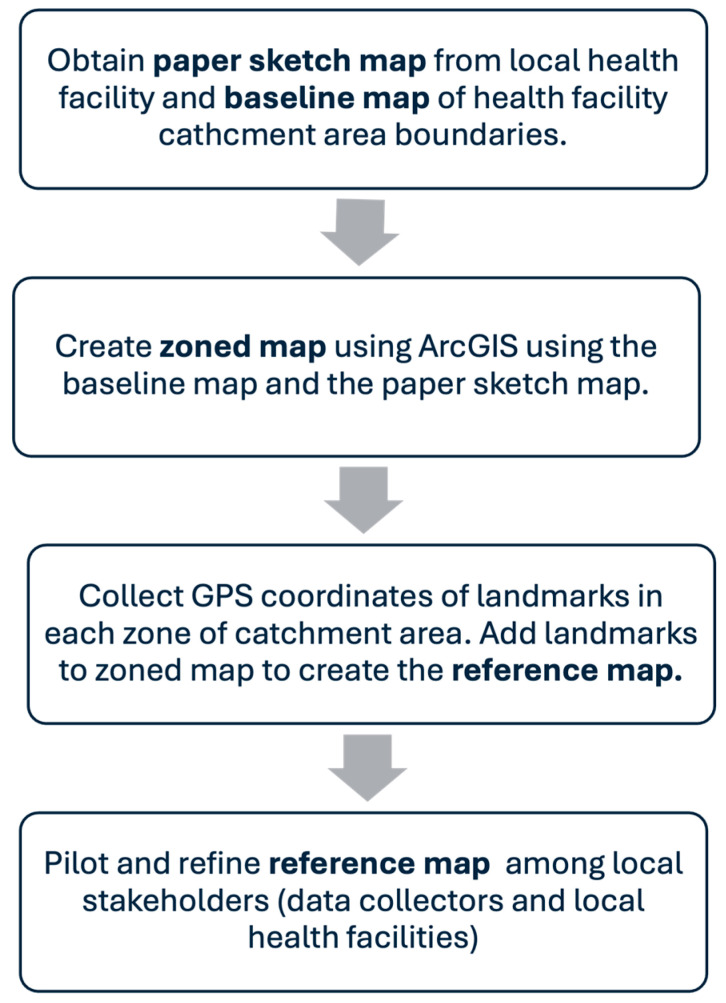
Reference map development process: A step-by-step approach to creating a reference map.

**Figure 2 vaccines-13-00924-f002:**
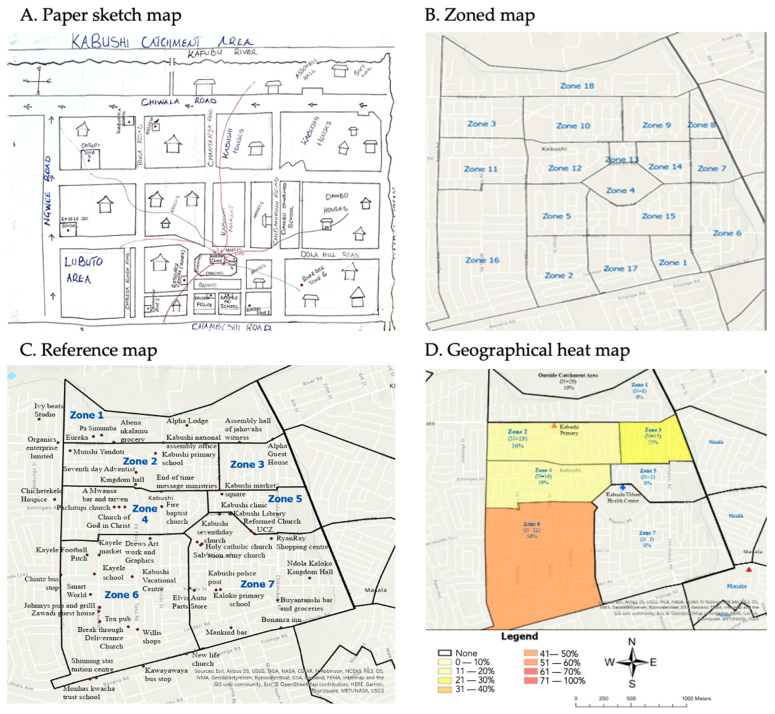
Mapping the Kabushi catchment area: The paper sketch map (**A**) provided by the health clinic and used to generate a zoned map (**B**). Prominent landmarks were then added to the zoned map to create the reference map (**C**), which was used during interviews. The geographical heat map (**D**) was developed using the school vaccination screening data.

**Figure 3 vaccines-13-00924-f003:**
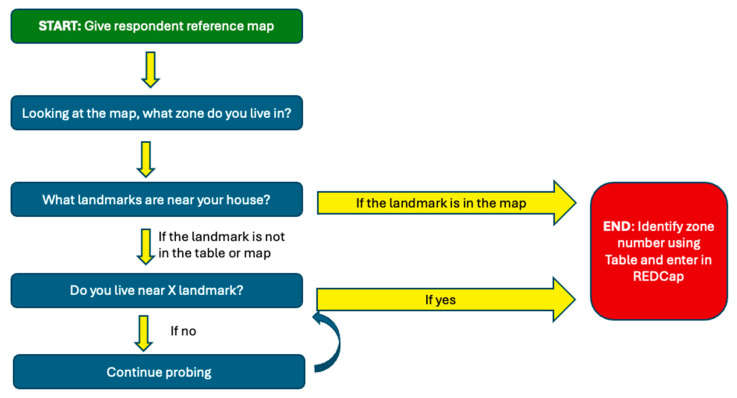
Probing questions flowchart: a visual guide to the data collector’s questioning process.

**Figure 4 vaccines-13-00924-f004:**
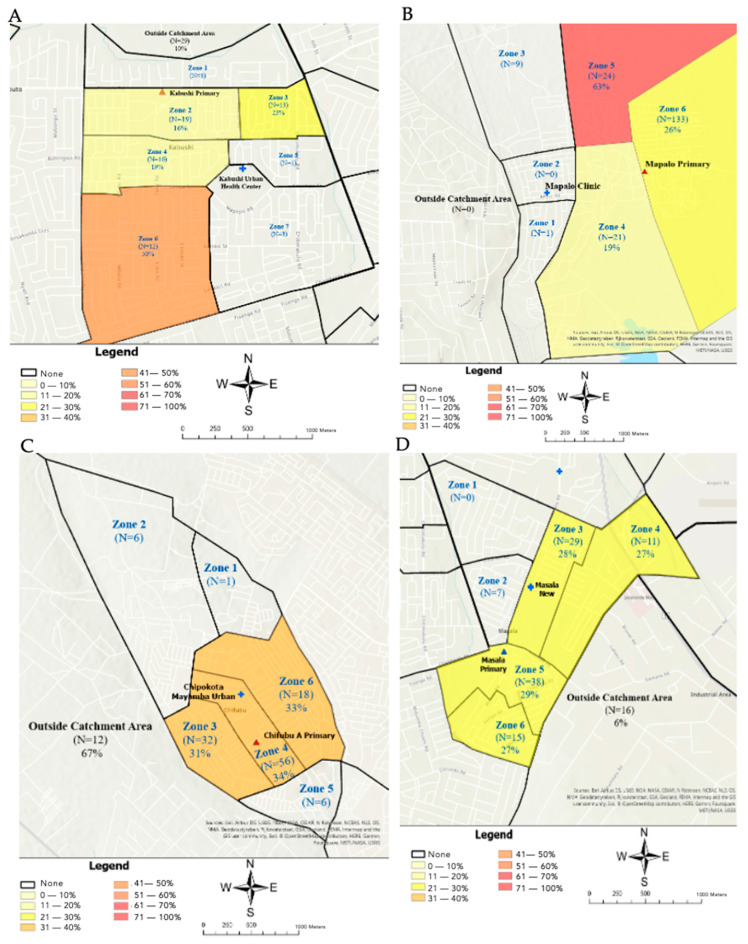
Geographic health maps: Vaccination status by health facility catchment area zone. (**A**) Proportion of under-vaccinated at Kabushi. (**B**) Proportion of under-vaccinated at Mapalo. (**C**) Proportion of under-vaccinated at Chifubu. (**D**) Proportion of under-vaccinated at Masala.

**Table 1 vaccines-13-00924-t001:** Percentage of under-vaccinated children by school.

	Masala	Kabushi	Chifubu	Mapalo	Total
Number of zones	6	7	6	6	25
Total children, *n*	116	100	131	188	535
Under-vaccinated, *n* (%)	32 (27.5)	15 (15)	48 (36.6)	56 (30.0)	151 (28.2)
Median children per zone	13	12	12	15	13
Range (min–max) per zone	0–38	1–19	1–56	0–133	0–133
Children living outside catchment area, *n*	16	29	12	0	57

**Table 2 vaccines-13-00924-t002:** Key lessons learned from developing the reference maps.

In a setting without street addresses, such as the setting where this project was implemented, conducting data collection outside of the household, such as a school or health facility, requires innovative approaches to collect geographic information for participants to approximate residential areas. Use of prominent and familiar landmarks on the maps allowed participants to identify which health facility catchment area zone the child lived in, facilitating an analysis of characteristics by zone to inform program planning. Successful GIS map development requires close collaboration with local communities and health facility staff, leveraging their knowledge to improve the map’s accuracy. The community must be the central driver in developing the baseline maps. Careful consideration is needed when developing boundaries for catchment areas and zones. More zones can increase precision but may reduce sample size, making it harder to compare characteristics (e.g., household, demographic or vaccination coverage) across zones. Consideration must be given to having a reasonable number of households in each zone and ensuring households falling along the boundaries are correctly assigned a zone. Repeated pilot testing and feedback from data collectors and community members are crucial for refining reference maps and ensuring they accurately reflect local realities. Extensive training for data collectors on using GIS maps and asking probing questions is vital to overcome challenges such as respondent illiteracy and to ensure accurate identification of the geographic distribution of individuals.

## Data Availability

Data will be made available on request after fulfilling the Zambia National Health Research Authority regulations and approvals.

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
