# Peer review of "School Entry Vaccination Checks Allow Mapping of Under-Vaccinated Children in Zambia"

_vaccines, 2025, doi:10.3390/vaccines13090924_

Round 1
Reviewer 1 Report
Comments and Suggestions for Authors
The paper addresses an important topic: undervaccination/underimmunization of children in LMICs. It is generally well-written, easy to follow, and methodologically sound. However, there are several points the authors should consider addressing. The introduction could be expanded and refined. The first paragraph, which introduces the topic, is too brief. It would benefit from stating the magnitude of the burden of vaccine-preventable diseases in LMICs, including data on the number of deaths averted by vaccination, and from discussing the various dimensions and determinants of undervaccination in greater depth. It would also be valuable to include a section specifically focused on Zambia, covering the local and national burden of communicable diseases, vaccination coverage rates, and relevant policy or programmatic context. Figure 2 is difficult to read and should be improved for clarity. It appears that the authors performed geospatial data visualization, but it is unclear whether any geospatial statistical analyses, such as geographically weighted regression or similar methods, were attempted, and clarification would be helpful. Finally, while the strengths and limitations section of the discussion is well-presented, the comparison with existing literature could be expanded to position the study’s findings more clearly within the broader research landscape.
Author Response
Comment 1: The paper addresses an important topic: undervaccination/underimmunization of children in LMICs. It is generally well-written, easy to follow, and methodologically sound. However, there are several points the authors should consider addressing. The introduction could be expanded and refined. The first paragraph, which introduces the topic, is too brief. It would benefit from stating the magnitude of the burden of vaccine-preventable diseases in LMICs, including data on the number of deaths averted by vaccination, and from discussing the various dimensions and determinants of undervaccination in greater depth. It would also be valuable to include a section specifically focused on Zambia, covering the local and national burden of communicable diseases, vaccination coverage rates, and relevant policy or programmatic context.
Response 1: As suggested, we revised the first paragraph to add more information on the value, gaps in vaccination, burden in VPDs and Zambia specific details. We also shifted around the introduction section so that this section flows better.
Comment 2: Figure 2 is difficult to read and should be improved for clarity. It appears that the authors performed geospatial data visualization, but it is unclear whether any geospatial statistical analyses, such as geographically weighted regression or similar methods, were attempted, and clarification would be helpful.
Response 2: We have improved Figure 2 and enlarged each of the images for clarity and updated the caption and also reformatted it. We refrained from performing geospatial statistical analyses or regressions due to our small sample sizes, especially within each zone. We have added this to the limitations section (lines 383-386).
Comment 3: Finally, while the strengths and limitations section of the discussion is well-presented, the comparison with existing literature could be expanded to position the study’s findings more clearly within the broader research landscape.
Response 3: We have expanded our comparison with existing literature to better position the lessons learned from our study in the discussion section (lines 325-337).
Reviewer 2 Report
Comments and Suggestions for Authors
Dear authors,
Thank you for submitting your contribution. The manuscript is well written and the methodology is described in detail. However, there are some areas that would require further investigation to strengthen the validity and potential scalability of the results. More specific comments are set out below
Main Comments
- The most significant limitation of the study is its design, which is based on children enrolled in five specially selected primary schools. The authors correctly acknowledge that this approach is not capable of capturing out-of-school children. The discussion should address more directly how this selection bias might affect outcomes. It is therefore suggested that this limitation be emphasized more in the abstract and discussion.
-
The study defines children with an unknown vaccination status as "unvaccinated" for analysis purposes. Although it represents a conservative approach from a public health perspective (it is better to assume that a child needs a vaccine when in doubt), it introduces a potential measurement bias. This may overestimate under-vaccination rates. Therefore it is suggested to:
(a) Provide a more robust justification for this decision in the text.
(b) Perform a sensitivity analysis in which cases with unknown status are excluded from the analysis, to see if the identified geographic clusters remain significant. This would significantly strengthen the robustness of the results. - The article concludes that the method is a "scalable and cost-effective approach" and a "more scalable alternative" to home visits. However, the study does not provide data on the costs or resources required to implement this pilot programme. The described process, which includes digitization of maps, field visits to collect GPS coordinates of landmarks, and "in-depth training for data collectors," suggests a significant initial investment of time and expertise. Authors are recommended to to provide a qualitative discussion (or quantitative data, if available) on the resources needed for implementation, addressing the challenges related to the cost of GIS technology and the need for local expertise.
- Figure 4 is central to the results of the study. In panels 4A and 4B, the sub-vaccination clusters are clear. In 4C and 4D panels, the distribution appears more uniform. The discussion could benefit from an analysis as to why some areas show clear clusters while others do not. Could it be related to sample sizes by zone or to real socio-demographic differences?
In general ,the manuscript needs more references (if available) to enhance and support your contribution
Author Response
Comment 1:
- The most significant limitation of the study is its design, which is based on children enrolled in five specially selected primary schools. The authors correctly acknowledge that this approach is not capable of capturing out-of-school children. The discussion should address more directly how this selection bias might affect outcomes. It is therefore suggested that this limitation be emphasized more in the abstract and discussion.
Response 1: We included the limitations with study design (i.e., reliance on 5 purposively select primary school and issues with representation) and selection bias in the abstract (lines 39-42) and discussion (lines 358-363).
Comment 2:
- The study defines children with an unknown vaccination status as "unvaccinated" for analysis purposes. Although it represents a conservative approach from a public health perspective (it is better to assume that a child needs a vaccine when in doubt), it introduces a potential measurement bias. This may overestimate under-vaccination rates. Therefore it is suggested to:
(a) Provide a more robust justification for this decision in the text.
(b) Perform a sensitivity analysis in which cases with unknown status are excluded from the analysis, to see if the identified geographic clusters remain significant. This would significantly strengthen the robustness of the results.
Response 2: To clarify, a child who did not receive a dose or had unknown status was marked as “no, dose not received” by teachers and nurses. The reason for treating those with unknown status as missing doses was to ensure that no child potentially missing doses was missed by the screening system, in line with public health practice. We have clarified this in the methods text (lines 131-134, 165-169). In our data set, vaccination status for each dose was recorded as “yes” or “no”, so it is not possible to disaggregate those with unknown status to conduct the sensitivity analysis.
Comment 3:
- The article concludes that the method is a "scalable and cost-effective approach" and a "more scalable alternative" to home visits. However, the study does not provide data on the costs or resources required to implement this pilot programme. The described process, which includes digitization of maps, field visits to collect GPS coordinates of landmarks, and "in-depth training for data collectors," suggests a significant initial investment of time and expertise. Authors are recommended to to provide a qualitative discussion (or quantitative data, if available) on the resources needed for implementation, addressing the challenges related to the cost of GIS technology and the need for local expertise.
Response 3: We have expanded the discussion on costs in the limitations citing available literature (lines 338-348) and have de-emphasized GIS as a cost-effective approach in the abstract and throughout, as this will depend on initial investments and local capacity building.
Comment 4:
- Figure 4 is central to the results of the study. In panels 4A and 4B, the sub-vaccination clusters are clear. In 4C and 4D panels, the distribution appears more uniform. The discussion could benefit from an analysis as to why some areas show clear clusters while others do not. Could it be related to sample sizes by zone or to real socio-demographic differences?
Response 4: As suggested, we highlighted the differences in the spatial distributions to the discussion (lines 279-289). We chose not to speculate the reason as to why some areas show clustering and others did not due to the small sample sizes within zones (noted in limitations line 385 ).
Comment 5:
In general ,the manuscript needs more references (if available) to enhance and support
Response 5: We have carefully reviewed the manuscript and added additional references where available and appropriate to strengthen and support our arguments.
Round 2
Reviewer 1 Report
Comments and Suggestions for Authors
The paper has considerably improved.